# Claudin-10 Expression Is Increased in Endometriosis and Adenomyosis and Mislocalized in Ectopic Endometriosis

**DOI:** 10.3390/diagnostics12112848

**Published:** 2022-11-17

**Authors:** Anna C. Löffelmann, Alena Hoerscher, Muhammad A. Riaz, Felix Zeppernick, Ivo Meinhold-Heerlein, Lutz Konrad

**Affiliations:** Department of Gynecology and Obstetrics, University of Giessen, 35392 Giessen, Germany

**Keywords:** endometrium, endometriosis, adenomyosis, claudin-10

## Abstract

Claudins, as the major components of tight junctions, are crucial for epithelial cell-to-cell contacts. Recently, we showed that in endometriosis, the endometrial epithelial phenotype is highly conserved, with only minor alterations. For example, claudin-11 is strongly expressed; however, its localization in the endometriotic epithelial cells was impaired. In order to better understand the role of claudins in endometrial cell-to-cell contacts, we analyzed the tissue expression and localization of claudin-10 by immunohistochemistry analysis and two scoring systems. We used human tissue samples (n = 151) from the endometrium, endometriosis, and adenomyosis. We found a high abundance of claudin-10 in nearly all the endometrial (98%), endometriotic (98–99%), and adenomyotic (90–97%) glands, but no cycle-specific differences and no differences in the claudin-10 positive endometrial glands between cases with and without endometriosis. A significantly higher expression of claudin-10 was evident in the ectopic endometrium of deep-infiltrating (*p* < 0.01) and ovarian endometriosis (*p* < 0.001) and in adenomyosis in the cases with endometriosis (*p* ≤ 0.05). Interestingly, we observed a shift in claudin-10 from a predominant apical localization in the eutopic endometrium to a more pronounced basal/cytoplasmic localization in the ectopic endometria of all three endometriotic entities but not in adenomyosis. Significantly, despite the impaired endometriotic localization of claudin-10, the epithelial phenotype was retained. The significant differences in claudin-10 localization between the three endometriotic entities and adenomyosis, in conjunction with endometriosis, suggest that most of the aberrations occur after implantation and not before. The high similarity between the claudin-10 patterns in the eutopic endometrial and adenomyotic glands supports our recent conclusions that the endometrium is the main source of endometriosis and adenomyosis.

## 1. Introduction

The presence of endometrial glands and stroma in the myometrium characterizes adenomyosis [1], whereas endometrial implants in the peritoneum, ovaries, and other loci are typical for endometriosis [2]. Retrograde menstruation causing the spreading of endometrial tissue, followed by its implantation on different surfaces in the pelvic or abdominal cavity, is generally accepted as the main cause of endometriosis [2,3]. However, despite a high rate (76% to 90%) of retrograde menstruation [4,5,6], only approximately 0.7–8.6% of women in the general population acquire endometriosis [7].

It was suggested that peritoneal endometriosis, endometriomas, and deep-infiltrating endometriosis (DIE) could represent three distinct entities, which do not share a common pathogenesis [8]. However, irrespective of their location, such as the ovary, peritoneum, or deep-infiltrating endometriosis, the ectopic endometriotic glands nearly always resemble histologically uterine endometrial glands [9].

Recently, the epithelial–mesenchymal transition (EMT) has been suggested to participate in the pathogenesis of endometriosis [10,11] and adenomyosis [12]. However, after an analysis of epithelial cell–cell contacts, mainly those of claudin-2, -3, -7, and -11 [13,14], and analysis of several EMT markers, we proposed that the persistence of epithelial cell–cell contacts indicates only a partial EMT in endometriosis, without the transition of epithelial cells into mesenchymal cells [15].

Claudins, as the major components of tight junctions (TJ), are mostly located in the apicolateral membranes of epithelial and endothelial cells and are often impaired in human cancer, thus permitting the escape of cancer cells and the acquisition of invasive properties [16]. In particular, a barrier function and strand formation are mainly attributed to the claudins. Within the TJ complex, it is predominantly the claudin composition that determines the properties of the epithelia, such as barrier-forming claudins (claudin-1, -3, -5, -11, -14, and -18), mainly in tight epithelia, or channel-forming claudins (claudin-2, -4, -10, -15, -17, -21), mediating the charge and size selectivity for the paracellular pathway [17]. Paracellular permeability can be fine-tuned according to the needs of a tissue by inserting these channel-forming claudins into TJs [18]. Thus, mutations in claudin-10b, a channel-forming claudin, cause severe diseases, such as the HELIX syndrome (hypohidrosis, electrolyte imbalance, lacrimal gland dysfunction, ichthyosis, and xerostomia), or a homozygous missense mutation results in anhidrosis severe heat intolerance and mild kidney failure [18,19]. A tissue-specific claudin-10 knockout in the thick ascending limb of mouse kidneys demonstrated hypermagnesemia, nephrocalcinosis, and moderate polyuria and polydipsia caused by reduced Ca^2+^ and Mg^2+^ excretion and increased Na^+^ loss [20,21]. Significantly, two alternative exons 1 in the claudin-10 gene encode two major isoforms, claudin-10a and claudin-10b, resulting in a different first transmembrane region and part of the first extracellular segment. In contrast to claudin-10b, which is ubiquitously expressed, claudin-10a expression seems to be restricted to the kidneys [22]. In contrast to claudin-10a, which is selective for anions, claudin-2, -10b, -15, and -21 are selective for cations, and claudin-4 and -17 are selective for anions and cations [17].

In the human endometrium and in endometriosis, several claudins, such as claudin-1-5, -7, and -11 [13,14,23,24,25], have been identified and characterized. In endometriotic lesions, claudin-3 and -7 have been found to be downregulated, whereas claudin-5 mRNA was decreased, but the protein levels were increased compared to the controls [23]. In the eutopic and ectopic endometrium, a highly similar localization of claudins-1-4 was found [14,23]. In contrast, a down-regulated expression of claudin-3 and claudin-4 in the ectopic endometrium, compared to the controls, was described [24].

In the endometrium, claudin-10 was found to be localized in the murine and human glandular epithelium, with a possible role in decidual angiogenesis and the regulation of trophoblast invasion [26]. Recently, an association was discovered between IL-22 deficiency and a decreased expression of claudin-2 and claudin-10 in endometrial regeneration after inflammation-triggered abortion in mice [27]. However, claudin-10 localization has never been analyzed in endometriosis or adenomyosis. In order to better understand the possible contribution of claudin-10 to the endometrial epithelial phenotype and its role in the pathogenesis of adenomyosis and endometriosis, we analyzed claudin-10 localization based through immunohistochemistry in this study.

## 2. Materials and Methods

### 2.1. Patients

This study was approved by the Ethics Committee of the Medical Faculty of the Justus-Liebig-University, Giessen, Germany (registry number 95/09). All the participants gave written informed consent. All the specimens were obtained by hysterectomy (uteri, n = 85) or laparoscopy (endometriotic tissues, n = 66 patients with 67 lesions) from patients mainly affected by pain or infertility (Table 1). The intraoperative findings were classified according to the revised American Society for Reproductive Medicine score (rASRM) and ENZIAN score in cases of DIE [28]. The dating of the endometrial tissue was based on its histological evaluation by the pathologist and the last menstrual period as reported by the patients.

The specimens were fixed in Bouin’s solution (and, partly, in formaldehyde for the histological evaluation by the pathologist) and embedded in paraffin. After staining 5 µm sections with hematoxylin and eosin, the histological evaluation was performed.

### 2.2. Immunohistochemical Analysis and Quantification

Serial sections of 5 µm were cut to ensure that, in most cases, the same lesions could be examined. The immunohistochemistry (IHC) of the Bouin-fixed specimens was performed as published recently [29]. The EnVision Plus System (DAKO, Hamburg, Germany) was used according to the manufacturer’s instructions. Briefly, antigen retrieval was performed with a citrate buffer (pH 6, DAKO), and then the jars containing the slides were placed in a steamer (Braun, Multi Gourmet) at 100 °C for 20 min and remained in the steamer for 20 min to cool. Primary antibodies against claudin-10 (diluted 1:100; Thermo Fisher, Life Technologies, cat no. 38-8400) were added, and incubation was performed in a humidified chamber overnight at 4 °C. After washing with PBS, incubation with the appropriate secondary antibody (anti-rabbit-labelled polymer with horse radish peroxidase, DAKO, cat no. K4003) was performed for 30 min at room temperature. The staining was visualized with diaminobenzidine (Liquid DAB K3467, DAKO). The counterstaining was performed with Mayer’s hematoxylin (Waldeck, Germany), and after their dehydration in ethanol, slides were mounted with Eukitt. Negative controls for IHC were prepared by replacement of the primary antibody with an IgG isotype (diluted 1:2000, Invitrogen, cat no. 02-6102) at the same concentration as the primary antibody.

Digital images were obtained with a Leica DM 2000/Leica MC170/Leica application suite, LAS 4.9.0, and then processed with Adobe Photoshop CS6. The IHC quantification was performed through use of the histological score (HScore: 0, no staining; 1+, weak but detectable; 2+, moderate or distinct; 3+, intense), which was calculated for each tissue by summing the percentages of the cells grouped in one intensity category and multiplying this number with the intensity of the staining. All the glands or cysts were used for the evaluation of the HScore. The quantification of the claudin-10 localization was performed using the following values: 5 for mainly apical, 4 for mainly apicolateral, 3 for mainly basolateral, 2 for mainly basal, 1 for mainly cytoplasmic, and 0 for no staining.

### 2.3. Statistics

All the values are presented as means ± standard error of the mean (SEM) or standard deviation (SD). The HScore values of the different groups were analyzed using one-way analysis of variance (ANOVA). Then, a comparison between two groups was conducted through the non-parametric test of Mann–Whitney, and comparisons between more than two groups were conducted using the test of Kruskal–Wallis. Here, *p*-values of ≤0.05 were considered to be significant. GraphPad Prism 6.01 (www.graphpad.com, accessed on 25 August 2020) was used for the statistics. The sample size was calculated with the following formula: samples size = [z^2^ SD(1-SD)]/ME^2^ (z = 1.96 for a confidence interval of 95%; SD = standard deviation of 0.5; and EM = error margin of 0.1), as given at www.qualtrics.com (accessed on 25 October 2022). A sample size of n = 96 was deemed sufficient for a confidence interval of 95% (50% standard deviation and 10% error margin). We used a sample size of n = 151.

## 3. Results

In this observational study, the IHC analysis of claudin-10 in patients with and without endometriosis showed a strong and mainly apical localization in nearly all the glands as well as in nearly all the luminal and glandular epithelial cells of the eutopic endometrium in the patients with and without endometriosis (Figure 1). However, currently, no antibodies are available for IHC on paraffin-embedded tissue to distinguish between claudin-10a and -10b. The quantification of claudin-10 using the HScore demonstrated a high degree of similarity between the patients with and without endometriosis, as well as the proliferative and secretory phases (Table 2). Similarly, claudin-10 was also found mainly in the apical cell poles and in nearly all the glands, as well as in nearly all the epithelial cells of the eutopic endometrium in the cases with and without endometriosis and all the cases that presented with adenomyosis (Figure 2). However, the quantification of the percentage of claudin-10 positive glands showed a significant reduction in the number in the eutopic endometrium among the cases with endometriosis in conjunction with adenomyosis (Table 3). Furthermore, the HScore revealed a high similarity between the endometrial and adenomyotic glands in the cases with and without endometriosis, although there was a significant increase in the adenomyotic glands in the cases with endometriosis together with adenomyosis (Table 3).

A positive immunoreactivity of claudin-10 could also be identified in nearly all the endometriotic epithelial cells as well as in nearly all the endometriotic lesions of the three endometriotic entities: ovarian (Figure 3A), peritoneal (Figure 3B), and deep-infiltrating endometriosis (Figure 3C). As the HScores of the eutopic endometria with and without endometriosis showed no significant differences (Table 2), we merged both datasets for a comparison with the three endometriotic entities. The HScore was significantly increased in deep-infiltrating and peritoneal endometriosis and especially in ovarian endometriosis (Table 4). In contrast, the percentage of positive glands was very similar between the eutopic and ectopic endometria (Table 4).

Remarkably, we found an altered localization of claudin-10 in the ectopic endometrium compared to the preferential apical localization in the eutopic endometrium (Figure 4). All three endometriotic entities, but not adenomyosis, showed significantly different localization patterns (Figure 4, Table 5). In particular, in ovarian endometriosis, a shift towards a more basal or cytoplasmic localization (39% of the cases) was prominent compared to the eutopic endometrium (0%).

## 4. Discussion

In this study, we analyzed the expression and localization of claudin-10 in the epithelial cells of eutopic endometrial and adenomyotic glands and endometriotic lesions in the ovary, peritoneum, and further organs in the pelvic cavity of patients with DIE. Our results convincingly demonstrate that claudin-10 is expressed in nearly all the eutopic, ectopic endometrial, and adenomyotic glands, although we noted a reduction in the eutopic endometrium in the cases with endometriosis and adenomyosis. Furthermore, we also showed that the expression of claudin-10 was significantly higher in the ectopic endometrium, especially in ovarian endometriosis, as well as in adenomyosis in cases with coexisting endometriosis. Of note, we identified a shift in the localization from the apical membrane in the eutopic endometrium to a more pronounced basal/cytoplasmic localization in the ectopic endometrium, again preferentially in ovarian endometriosis.

Our results regarding claudin-10 in the endometrium are in line with the cycle-independent expression of claudin-1, 3, 4, 5, and 11 [13,23,24]. Additionally, claudin-10 was found in nearly all the epithelial cells and glands in the endometrium, although there was a significant reduction in the percentage of positive glands in the endometrium in the cases with endometriosis together with adenomyosis.

Recently, we suggested that the endometrial glands are the main source of adenomyotic glands because of the highly identical protein pattern in the endometrium compared to that in adenomyosis [30], which we similarly observed in the case of claudin-10. In another study, 3D reconstructions of the human endometrium demonstrated that the adenomyotic glands are still connected to the endometrial glands [31], supporting a pathogenesis model of adenomyosis, which seems to rely on cellular proliferation and invagination into the myometrium. This stands in clear contrast to the view that the eutopic endometrial cells of patients with and without endometriosis are clearly different [32]. The very high percentage of claudin-10 positive ectopic glands and the significantly increased HScores in the three endometriotic entities suggest that most of the distinct changes occur after implantation and not before. Similarly, we recently proposed that the partial EMT-like changes in the eutopic endometrium are relatively subtle and that the majority of differences can be observed after and not before implantation [15]. In agreement with prior studies [33,34], we too propose that these differences may be explained as a direct consequence of the different environments, such as the peritoneal fluid and the intraovarian microenvironment of the lesions, in relation to the intrauterine environment.

Interestingly, in our study, we identified a shift in the localization of claudin-10 from the apical membrane in the eutopic endometrium to a basal/cytoplasmic localization in the ectopic endometrium but not in adenomyosis. Similarly, in hepatocellular carcinoma (HCC) cells, claudin-10 was detected in the cytoplasm [35]. Additionally, for claudin-2, a cytoplasmic shift in endometrioid endometrial carcinoma was reported [25], as was also the case for claudin-11 in endometriosis [13]. Remarkably, only one mutation in the fourth transmembrane domain of claudin-10 was sufficient to cause a shift from the membrane to the cytoplasm and an impaired tight junction strand formation in the human kidney, resulting in a tubulopathy [36]. The question of whether mutations of claudin-10 are also responsible for the cytoplasmic localization in cancer cells or ectopic endometrial cells remains to be determined.

Remarkably, the highly similar cellular membrane localizations of claudin-10 in the adenomyotic epithelial cells compared to the endometrial cells are clearly different to the strongly impaired membrane localizations in the three endometriotic entities. Thus, increasing evidence supports the hypothesis of the difference between the pathogenesis of adenomyosis, which most probably occurs via invagination, and that of endometriosis, which relies on endometrial tissue breakdown, migration, and implantation into the ectopic sites, as already suggested by Sampson [3].

A higher expression of claudin-10 was also observed in several carcinomas, such as HCC [35], papillary thyroid cancer [37], and biliary tract cancers [38], and was correlated with a shorter overall survival in HCC [35]. Consequently, the down-regulation of claudin-10 in HCC was associated with a prolonged disease-free survival after a curative therapy [39]. Additionally, a knockdown of claudin-10 with small interfering RNA in a highly invasive HCC cell line abolished the invasion and strongly decreased the activation of matrix metalloproteinases and expression of claudins [39].

In addition to mediating cell–cell contacts, claudin-10 is also involved in regulating paracellular transport [19]. Calcium is one of the major signaling molecules that is required for the modulation of the cell cycle and apoptosis in oocytes [40]. A moderate Ca^2+^ increase mediates the spontaneous resumption of meiosis from diplotene, thus driving meiotic progression [41], whereas higher levels cause meiotic cell cycle arrest and apoptosis [40]. There is an intriguing possibility that the altered localization and increased expression of claudin-10, especially in ovarian endometriosis, might contribute to infertility via disturbed ion homeostasis.

The current study is based mainly on immunohistochemistry and, thus, has some limitations. Because of the scarcity of human tissues, it was not possible to analyze more samples. However, it is highly likely that the very high abundance and similarity of claudin-10 expression in the eutopic endometrium would be also observed with the use of more samples. Furthermore, the mRNA expression, protein abundance, migration, and invasion should be investigated using isolated endometrial and endometriotic stromal and epithelial cells to study the function of claudin-10 in endometrial TJs and in the disease pathogenesis in greater depth.

## 5. Conclusions

In this study, we identified similar high expressions of claudin-10 in nearly all the eutopic, ectopic endometrial, and adenomyotic glands, excepting a reduction in the eutopic endometrium among the cases with endometriosis and adenomyosis. The different expressions of claudin-10 in the ectopic compared to the eutopic endometrium, especially in ovarian endometriosis, suggest that most of the changes occur after and not before implantation. Despite these changes, the endometrial and endometriotic epithelial cells still retained their epithelial phenotype and intact cell-to-cell contacts at the ectopic sites. The shift in the localization of claudin-10 from the apical membrane in the eutopic endometrium to a more pronounced basal/cytoplasmic localization in ectopic endometrium might contribute to a disturbed ion homeostasis, which warrants further investigation.

## Figures and Tables

**Figure 1 diagnostics-12-02848-f001:**
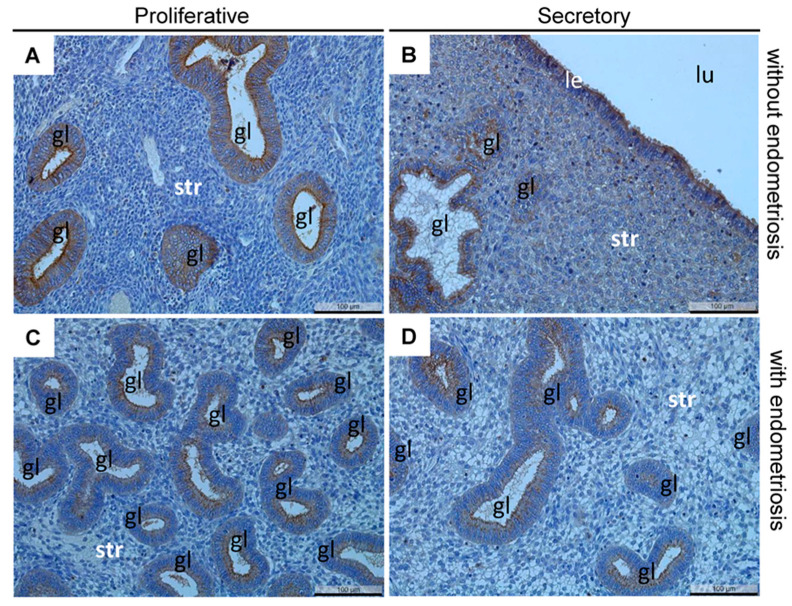
Representative microphotographs of claudin-10 in the proliferative (**A**) and secretory (**B**) endometrium without endometriosis and in the proliferative (**C**) and secretory (**D**) endometrium with endometriosis. Two patients also had leiomyomas (**A**,**C**). One patient had endometriosis in the uterosacral ligament (**C**), and one patient had endometriosis in the bladder (**D**). gl, gland; str, endometrial stroma; le, luminal epithelium; lu, lumen; scale bars 100 µm; magnification 200×.

**Figure 2 diagnostics-12-02848-f002:**
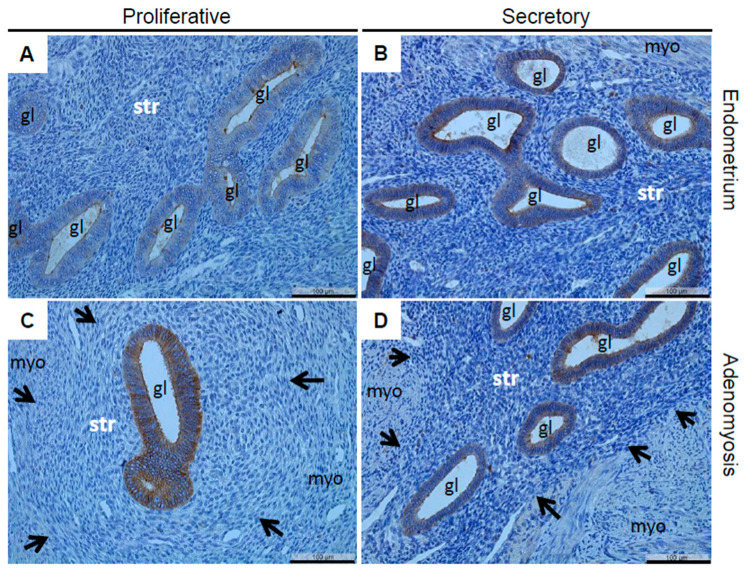
Representative microphotographs of claudin-10 in cases with matched endometrium and adenomyosis lesions. A proliferative endometrium (**A**) with the corresponding adenomyosis (**C**) and a secretory endometrium (**B**) with the corresponding adenomyosis (**D**) are presented. One patient also showed endometriosis in the vagina, rectum, and colon (**A**,**C**). The other patient presented with a leiomyoma (**B**,**D**). gl, gland; str, endometrial/adenomyotic stroma; myo, myometrium; arrows denote the boundaries of the adenomyotic lesions; scale bars 100 µm, magnification 200×.

**Figure 3 diagnostics-12-02848-f003:**
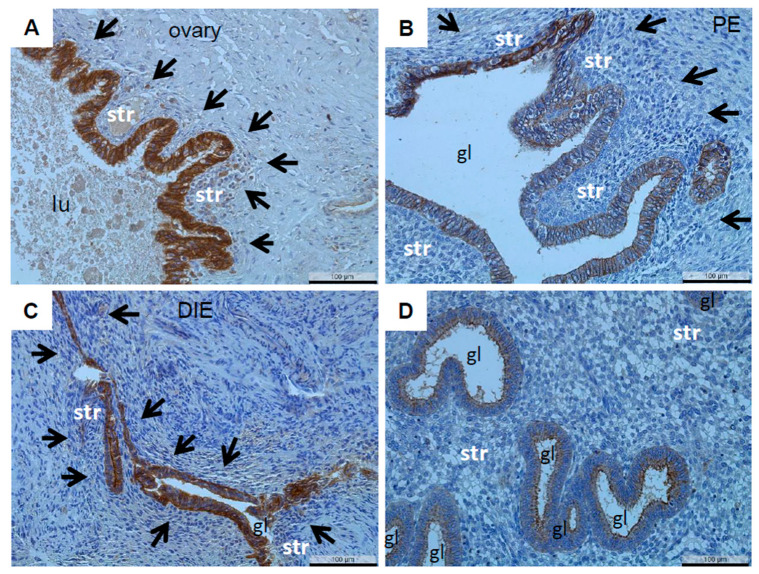
Representative microphotographs of claudin-10 in ovarian endometriosis (**A**), ovarian cyst), peritoneal endometriosis ((**B**), pouch of Douglas), and DIE ((**C**), bladder). For a better comparison, a proliferative endometrium is also shown (**D**). PE, peritoneal endometriosis; DIE, deep-infiltrating endometriosis. lu, lumen; gl, gland; str, endometrial stroma; arrows denote the boundaries of the lesions; scale bars 100 µm, magnification 200×.

**Figure 4 diagnostics-12-02848-f004:**
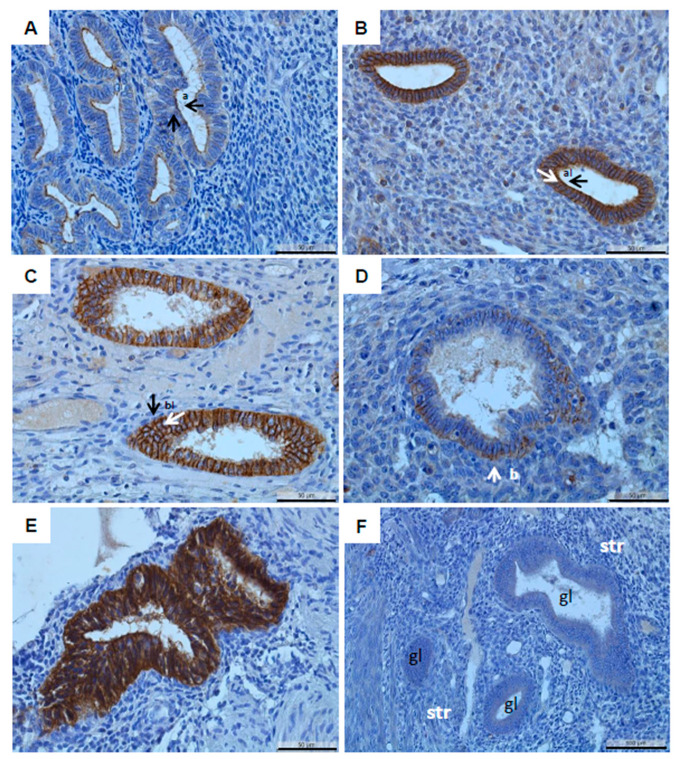
Representative microphotographs of claudin-10 showing the different localizations: apical ((**A**), a, score 5), apicolateral ((**B**), al, score 4), basolateral ((**C**), bl, score 3), mainly basal ((**D**), b, score 4), and cytoplasmic ((**E**), score 5). A representative negative control with the rabbit IgG isotype on the proliferative endometrium is provided (**F**). A secretory endometrium of a patient with leiomyoma (**A**), a secretory endometrium of a patient with endometriosis in the rectum (**B**), a secretory endometrium of a patient (**C**), a patient with peritoneal endometriosis in the utero-sacral ligament (**D**) and further lesions in the rectum and bladder, and a patient with DIE in the rectum (**E**) are shown. The arrows denote typical claudin-10 cellular localizations. Gl, gland; str, endometrial stroma; scale bars 50 µm (**A**–**E**) and 100 µm (**F**); magnification 400× (**A**–**E**) and 200× (**F**).

**Table 1 diagnostics-12-02848-t001:** Overview of the tissue samples.

Tissues	EM (ctrl)	EM-EN	EN-AM	AM	Ov-EM	Pe-EM	DIE
All samples (n)	23	24	17	21	27 (28)	18	21
Mean age	42 ± 8.0	37 ± 7.5	41 ± 4.4	44 ± 4.8	33 ± 6.7	35 ± 4.5	36 ± 6.8
Proliferative (n)	12	9	8	9			
Mean age	43 ± 9.6	32 ± 5.5	44 ± 3.9	43 ± 5.5			
Secretory (n)	11	15	9	12			
Median age	41 ± 6.1	39 ± 7.4	39 ± 3.5	44 ± 4.5			
Leiomyoma	16	9	8	11			
Bladder							3
Uterosacral lig.							4
Ovarian fossa						9	
Pouch of Douglas						1	
Round lig. of uterus						5	
Peritoneum						1	
Rectum						2	5
Pararectal fossa							2
Rectosigmoid							2
Paraurethral							1
Sigmoid colon							2
Colon							2

E.g., n = 19 (20) means 20 lesions from 19 patients; ctrl, control; lig, ligament; AM, adenomyosis; EM, endometrium; EN, endometriosis; Ov, ovarian; Pe, peritoneal; DIE, deep-infiltrating endometriosis. The median age is given with the SD.

**Table 2 diagnostics-12-02848-t002:** Characterization of claudin-10 in endometria with and without endometriosis according to the HScore and percentage of positive glands.

	EM without EN		EM with EN	
	P	S	P	S
N (median age)	12 (43)	11 (41)	9 (32)	15 (39)
HScore				
Mean	103	106	108	95
SEM	5.5	11.3	14.6	5.0
*p*-values	n.s.	n.s.	n.s.	n.s.
% Positive glands				
Mean	98	97	97	98
SEM	1.2	1.7	2.5	1.3
*p*-values	n.s.	n.s.	n.s.	n.s.

EM, endometrium; EN, endometriosis; SEM, standard error of the mean; n.s., not significant

**Table 3 diagnostics-12-02848-t003:** Quantification of claudin-10 in adenomyosis compared to endometria with and without endometriosis.

	EM (ctrl) EM-EN		EM-EN-AM		EM-AM	
	(a)	(b)	EM (c)	AM (d)	EM (e)	AM (f)
N	23	24	12	17	14	21
Age	42 ± 8.0	37 ± 7.5	41 ± 4.4	41 ± 4.4	43 ± 5.0	44 ± 4.8
HScore						
Mean	104	100b	107	136d	97	108
SEM	6.0	6.2	9.0	11.0	5.2	4.8
*p*-value	n.s.	0.05 ^b,d^	n.s	0. 05 ^b,d^	n.s.	n.s.
% Positive glands						
Mean	98	98	90	97	95	96
SEM	1.0	1.2	3.0	1.7	2.4	2.2
*p*-value	n.s.	0.05 ^b,c^	0.05 ^b,c^	0.05 ^c,d^	0.05 ^c,e^	n.s.
			0.05 ^c,d^			
			0.05 ^c,e^			

Ctrl, control group; EM, endometrium; EM-EN, endometrium with endometriosis; EM-EN-AM, endometrium with endometriosis and adenomyosis; EM-AM, endometrium with adenomyosis; n.s., not significant; e.g. b,c shows the statistical difference between column (**b**) and column (**c**); the same applies for c,d; c,e; and c,d. The age is given as the mean ± SEM.

**Table 4 diagnostics-12-02848-t004:** Comparison of eutopic and ectopic endometrial glands with claudin-10.

	EM-EN	EM (ctrl)	EM-EN-AM	Ov-EN	Pe-EN	DIE
	(a)	(b)	(c)	(d)	(e)	(f)
N	23	24	12	27 (28)	18	21
Age	42 ± 8.0	37 ± 7.5	41 ± 4.4	33 ± 6.7	35 ± 4.5	36 ± 6.8
HScore						
Mean	104	100	107	170	136	130
SEM	6.0	6.2	9.0	8.4	13.2	8.0
*p*-value	0.001 ^a,d^	0.001 ^b,d^	0.001 ^c,d^	0.001 ^a,d^	n.s.	0.05 ^b,f^
		0.05 ^b,f^		0.001 ^b,d^		
				0.001 ^c,d^		
% Positive glands						
Mean	98	98	90	99	98	99
SEM	1.0	1.2	3.0	0.4	1.4	0.5
*p*-value	0.05 ^a,c^	0.01 ^b,c^	0.001 ^c,d^	0.001 ^c,d^	0.001 ^c,e^	0.01 ^c,f^
			0.001 ^c,e^			
			0.01 ^c,f^			
			0.05 ^a,c^			
			0.01 ^b,c^			

Ctrl, control group; EM, endometrium; EM-EN, endometrium with endometriosis; EM-EN-AM, endometrium with endometriosis and adenomyosis; Ov-EN, ovarian endometriosis; Pe-EN, peritoneal endometriosis; DIE, deep-infiltrating endometriosis; n.s., not significant, e.g., a,c shows the statistical difference between column (**a**) and column (**c**), the same applies for b,c; c,d; c,e; and c,f. The age and scores are given as means ± SEM.

**Table 5 diagnostics-12-02848-t005:** Localization scores of claudin-10 in the eutopic and ectopic endometrial glands.

	EM (ctrl)	EM-EN	AM	Ov-EN	Pe-EN	DIE
	(a)	(b)	(c)	(d)	(e)	(f)
N	23	24	38	27 (28)	18	21
Age	42 ± 8.0	37 ± 7.5	43 ± 8.0	33 ± 6.7	35 ± 4.5	36 ± 6.8
Mean	4.52	4.58	4.61	2.75	3.39	3.14
SEM	0.14	0.15	0.12	0.26	0.42	0.33
*p*-value	0.001 ^a,d^	0.001 ^b,d^	0.001 ^c,d^	0.001 ^a,d^	0.01 ^b,f^	0.01 ^b,f^
	0.05 ^a,f^	0.01 ^b,f^	0.001 ^c,f^	0.001 ^b,d^		0.001 ^c,f^
				0.001 ^c,d^		

Ctrl, control group; EM, endometrium; EM-EN, endometrium with endometriosis; AM, adenomyosis; Ov-EN, ovarian endometriosis; Pe-EN, peritoneal endometriosis; DIE, deep-infiltrating endometriosis; n.s., not significant, e.g., a,d shows the statistical difference between column (**a**) and column (**d**), the same applies for a,f; b,d; b,f; c,d; c,f; and a,d. The age and score are given as means ± SEM.

## Data Availability

The original data can be obtained from the corresponding author upon request.

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
