# Peer review of "Claudin-10 Expression Is Increased in Endometriosis and Adenomyosis and Mislocalized in Ectopic Endometriosis"

_diagnostics, 2022, doi:10.3390/diagnostics12112848_

Round 1
Reviewer 1 Report
The manuscript is written well and easy to read. The overall strategy seems very interesting. Valid research question flaws with the methodology and interpretation of the results. The methods are properly and appropriate conducted. The results were analyzed and interpreted correctly. The conclusions drawn are fully supported by the data. The obtained evidence support strongly the authors conclusion. There are no ethical concerns or other issues I believe the Editor should be aware. I think, the study fully match the standards of the journal. Therefore, I strongly recommend to accept this manuscript for the publication.
Author Response
Thanks for your comments. No changes have been suggested
Reviewer 2 Report
In the introduction section it is important to note ameliorated quality and PMID:34718292.
Author Response
The reviewer suggested to include PMID: 34718292. However, although this review is interesting the topic of endometriosis, pain and mental health has nothing to do with our manuscript. We have improved the Introduction section especially to better define our aims.
Reviewer 3 Report
This is an observational study where authors aimed to analyse tissue expression and localization of claudin-10 in the human endometrium, endometriosis, and adenomyosis by immunohistochemistry. They concluded that there is a high expression of claudin-10 in nearly all eutopic, ectopic endometrial and adenomyotic glands, except for a reduction in eutopic endometrium of cases with endometriosis and adenomyosis. This is a very interesting study, with important concept. I think there are certain issues / points to be revisited. The abstract should contain all necessary sections. As it stands, it is a narrative description of the results section. In the introduction section, in its last paragraph, authors should more specifically give the gap and their rationale of the study, and its description of the methods they aim to use. There must be synchronization between all sections. Registration number in any database – if exist- could be added. My main concern is on the absence of sample size measurement. Sample size calculation using software or based on existing similar studies is necessary. In that way, eligibility criteria could be more strictly and accurately described. The final conclusions should reflect the quality of this particular study.
Author Response
We are thankful for the remarks and insightful comments of the reviewers. We have tried our best to answer all queries.
Reviewer 3:
Q1: The abstract should contain all necessary sections. As it stands, it is a narrative description of the results section.
Answer 1: The Abstract in the journal Diagnostics is unstructured. We have changed some sentences to better explain our aims and also the conclusions are rewritten.
Q2: In the introduction section, in its last paragraph, authors should more specifically give the gap and their rationale of the study, and its description of the methods they aim to use.
Answer 2: We have changes the last paragraph as suggested and hope that it has improved.
Q3: There must be synchronization between all sections. Registration number in any database – if exist- could be added.
Answer 3: Sorry, but we do not understand the meaning of synchronization between all sections. Further, we do not have a registration number for our scientific studies.
Q4: My main concern is on the absence of sample size measurement. Sample size calculation using software or based on existing similar studies is necessary. In that way, eligibility criteria could be more strictly and accurately described.
Answer 4: To the best of our knowledge only very few scientific studies with immunohistochemistry have ever provided a sample size calculation. Despite these shortcomings we have found a formula to calculate the sample size. For our study a sample size of n=96 would have been sufficient. We have used n=151 samples.
Q5: The final conclusions should reflect the quality of this particular study.
Answer 5: We have changed the conclusion section and hope that it now better reflects the quality of our study.
Round 2
Reviewer 3 Report
There has been a fair effort towards the improvement of the reporting of this paper.
I do understand that the abstract should be unstructured. But I also know that it should contain all necessary sections of a structured one.
As for the last paragraph of the introduction section, I would suggest authors to focus on claudin 10, which is the main substance the study refers to, and connect it with the current literature – its gap_. Perhaps, some sentences could be removed.
Synchronisation means what authors are reporting in the materiasl and methods section to be paralleled with the exact reporting in the results and discussion sections.
The sample size calculation is fine.
The quality of the study is that of an observational one; in this respect, authors could make some small changes in the final results. In the conclusions section, the possible potential links could be moved to the discussion section.
Author Response
We are thankful for the remarks and insightful comments of the reviewer. We have tried our best to answer all queries.
Reviewer 3:
Q1: I do understand that the abstract should be unstructured. But I also know that it should contain all necessary sections of a structured one.
Answer 1: We have changed the Abstract that it now better resembles a structured one.
Q2: As for the last paragraph of the introduction section, I would suggest authors to focus on claudin 10, which is the main substance in the study refers to, and connect it with the current literature – its gap_. Perhaps some sentences could be removed.
Answer 2: We have changed the last paragraph as suggested and hope that it has improved.
Q3: Synchronization means what authors are reporting in the material and methods section to be paralleled with the exact reporting in the results and discussion sections.
Answer 3: We have now extensively rewritten the Discussion section that it now parallels the Results section.
Q4: The quality of the study is that of an observational one; in this respect, authors could make some small changes in the final results. In the conclusions section, the possible potential links could be moved to the discussion section.
Answer 4: We have made small changes as suggested, also shortened the conclusions, and have discussed some issues more clearly.